# Amino Anthraquinone: Synthesis, Characterization, and Its Application as an Active Material in Environmental Sensors

**DOI:** 10.3390/ma13040960

**Published:** 2020-02-21

**Authors:** Salman Ali, Muhammad Tahir, Nasir Mehboob, Fazal Wahab, Steven J. Langford, Suhana Mohd Said, Mahidur R. Sarker, Sabariah Julai, Sawal Hamid Md Ali

**Affiliations:** 1Faculty of Science, Engineering and Technology, Swinburne University of Technology, Hawthorn Victoria 3122, Australia; salmansafi92@gmail.com; 2Department of Physics, Faculty of Physical and Numerical Sciences, Abdul Wali Khan University Mardan, Khyber Pakhtunkhwa 23200, Pakistan; 3Department of Electrical Engineering, Faculty of Engineering, University of Malaya, Kuala Lumpur 50603, Malaysia; smsaid@um.edu.my; 4Department of Physics, Riphah International University, Islamabad 45210, Pakistan; nasir.mehboob@riphah.edu.pk; 5Department of Physics, Karakoram International University, Gilgit 15100, Pakistan; fazal.wahab@kiu.edu.pk; 6Department of Chemistry and Biotechnology, Swinburne University of Technology, Hawthorn Victoria 3122, Australia; sjlangford@swin.edu.au; 7Department of Mechanical Engineering, Faculty of Engineering, University of Malaya, Kuala Lumpur 50603, Malaysia; mahidur@um.edu.my (M.R.S.); sabsz@um.edu.my (S.J.); 8Department of Electric, Electronics and System Engineering, Faculty of Engineering and Built Environment, Universiti Kebangsaan Malaysia, Bangi 43600, Malaysia; sawal@ukm.edu.my

**Keywords:** humidity sensor, temperature sensor, 2-aminoanthraquinone (AAq), organic semiconductor sensor, AFM

## Abstract

This work reports synthesis, thin film characterizations, and study of an organic semiconductor 2-aminoanthraquinone (AAq) for humidity and temperature sensing applications. The morphological and phase studies of AAq thin films are carried out by scanning electron microscope (SEM), atomic force microscope (AFM), and X-ray diffraction (XRD) analysis. To study the sensing properties of AAq, a surface type Au/AAq/Au sensor is fabricated by thermally depositing a 60 nm layer of AAq at a pressure of ~10^−5^ mbar on a pre-patterned gold (Au) electrodes with inter-electrode gap of 45 µm. To measure sensing capability of the Au/AAq/Au device, the variations in its capacitance and resistance are studied as a function of humidity and temperature. The Au/AAq/Au device measures and exhibits a linear change in capacitance and resistance when relative humidity (%RH) and temperature are varied. The AAq is a hydrophobic material which makes it one of the best candidates to be used as an active material in humidity sensors; on the other hand, its high melting point (575 K) is another appealing property that enables it for its potential applications in temperature sensors.

## 1. Introduction

Organic semiconductors are revolutionizing the way we manufacture and engage with electronic devices as they enable the fabrication of light-weight, flexible electronic devices with novel form factors [1,2,3]. As opposed to inorganic semiconductors, such as crystalline silicon, organic semiconductors also have the ability to be processed at low temperatures and are easily deposited over large areas [4]. Organic semiconductors are, as the name suggests, materials which have electronic properties of metallic semiconductors along with the chemical and mechanical benefits of organic materials [5]. Their commercial use was in demonstrated in junction diodes, rechargeable batteries [6], organic solar cells [7], organic light emitting diodes (OLED) [8], organic field effect transistors (OFET) [9], and sensors [10].

Many of the organic semiconductors known are very stable and sensitive towards temperature, radiation and humidity [11,12,13]. Low molecular weight organic semiconductors-based capacitive and resistive surface type sensors are of great interest due to their low cost and simple fabrication methods. As a result, of their strong π-conjugated system and small molecular weight, organic semiconducting materials like perylene were shown to have very favorable characteristics for fabrication of organic devices such as organic sensors [14] and organic solar cells [15]. Many perylene derivatives have high melting points and are hydrophobic, which make them favorable for sensing of temperature and humidity [14]. In the visible spectrum, perylene absorbs strongly [15] and is both chemically and thermally very stable [16]. Perylene, as an n-type material [17], is a strong candidate for the fabrication of organic devices such light emitting diodes (LED) [16], field effect transistors (FET) [18] and solar cells [19]. 

Humidity and temperature sensors based on inorganic semiconductors are dominantly available in the market due to their stability, long lifespan, and reliability. However, the inorganic semiconductors are expensive, brittle, and high temperature fabrication techniques to be processed inside cleanrooms. On the other hand, organic semiconductors are the alternate materials which are low cost, mostly solution processable, mechanically flexible, light-weight, comparatively low-temperature processed and require simple fabrication techniques for device formation [20]. The faster response and recovery time of organic semiconductor-based sensors make them more attractive than other sensors [21]. Humidity and temperature sensors using organic semiconductors are commonly separated in two classes: capacitive or resistive. Due to low power consumption, capacitive-type humidity sensors are more popular [22] since low power sensors are very attractive to reduce the power consumption so that they can be powered by energy harvesting system [23,24,25]. In comparison with resistive-type sensors, capacitive-type sensors show better stability at higher humidity level and temperature [26,27,28,29]. Most sensors based on organic semiconductors are fabricated in either sandwich or surface type configurations with the former deemed reliable but suffer being easily damaged are susceptible to shortening problems [30]. Surface type sensors are easy to fabricate, are lower cost alternatives and offer simple technology for exploring different properties of organic semiconductors, such as the effect of humidity [29], the effect of temperature [31] and light sensitivity [11]. Various surface type resistive and capacitive sensors using the organic semiconductors, methyl orange (MO) [32], perylenes [13], porphyrins [33] and phthalocyanines [10,34] were previously reported. However, in the aforementioned semiconductor-based sensors, the detection of change in capacitance in response to the humidity of was observed from around 50% or 60%RH which meant that below 50%RH the sensors were inactive, whereas a good sensor suitable for practical applications should have the quality to sense the in broad range. Therefore, a quest for new functional materials is continued which could have a faster response with broad range of sensitivity and stability. For this reason, 2-aminoanthraquinone (AAq) is synthesized to explore its potential for the desired applications.

In this work, we fabricate surface type humidity and temperature sensors of the form (Au/AAq/Au) using 2-aminoanthraquinone (AAq) as the organic semiconductor for the first time due to its hydrophobic nature. The resistive and capacitive responses of the fabricated sensors were calculated as a function of relative humidity and temperature under ambient conditions. As the sensitivity of the sensor is frequency dependent, this device was characterized at both 120 Hz and 1 kHz. 

## 2. Experimental Work

### 2.1. Synthesis of 2-Amino Anthraquinone

To synthesize the 2-amino anthraquinone, we used the method mentioned in [35]. Briefly, we added sodium anthraquinone-2-sulfonate (20 g) and concentrated aqueous ammonia (200 mL, 0.88 g/mL). The mixture was heated in autoclave up to 180 °C and was maintained for 6 h at this temperature. The autoclave was allowed to cool down overnight and, hence, 2-aminoanthraquinone was filtered off and dried. The product was solid powder (red color, m.p. 302 °C). The reaction is schematically shown in Figure 1.

### 2.2. Device and Thin Film Preparation

The general device structure is shown in Figure 2. Commercially available glass slides were chosen as the substrate. Before deposition, the substrate was cleaned in acetone for 10 min followed by isopropanol for further 10 min using an ultrasonic bath (Elma Elmasonic P300H, Germany). The substrate was further cleaned by producing plasma inside the Edward Auto 306 thermal evaporator (West Union, SC, USA). Au electrodes of 50 nm thickness were deposited side by side and a 45 μm gap was created between electrodes, using a mask. After deposition, a 60 nm thick film of AAq was deposited at ~10^−5^ mbar base pressure to fabricate the Au/AAq/Au device. The experiments used an in situ FTM5 quartz crystal thickness monitor for film thickness measurements. 

### 2.3. Film and Device Characterization

XRD measurements were achieved using a Bruker Advance X-ray solutions D8 Diffractometer (Durham, UK) with CuKα radiation (λ = 0.1542 nm) source. Morphological studies were achieved using a Phillips XL 30 SEM (North Billerica, MA, USA) system and Nano-surf 3000 controllers (Langen, Germany) for AFM. The resistive and capacitive measurements as a function of relative humidity were carried out by placing the devices in a sealed glass chamber in which humidity is controlled through synthetic air. The resistive and capacitive measurements as a function of temperature were carried out by placing the devices on hotplate to increase the temperature in air at atmospheric pressure. The resistive and capacitive responses were recorded using an ESCORT ELC-132A LCR meter (Taipei, Taiwan) and the humidity measured using a SERIE- P320 standard humidity meter (China) as shown in Figure 3a,b shows the temperature measurement rig.

## 3. Results and Discussion

### 3.1. Materials Characterization

The crystalline or amorphous nature of thin films of AAq was investigated using X-ray diffraction techniques Figure 4. The X-ray diffraction pattern of AAq shows little structure with a single broad diffuse peak indicative of the amorphous nature of the thin film on glass. 

The SEM images of AAq thin film are shown in Figure 5a while the agglomerates in the film are shown at higher resolution in Figure 5b. It is clear from the SEM images that the AAq thin film surface and agglomerates in it are not uniform but rather rougher in its morphology. The voids formed and roughness are expected to be favorable for sensing properties, as these pores absorb water vapors while roughness trap the water vapors resulting an increase the effective dielectric constant of the film and hence increase the capacitance response vs. humidity.

The surface morphology of AAq thin films was also studied by AFM (Figure 6). The surface analysis of glass substrate and height and phase images of AAq thin film also shows a non-uniform rough surface at a more microscopic level. The average roughness (Ra) of the thin film on glass substrate is 1.470 nm. The root-mean square (RMS) value of the film roughness is 3.457 nm. Irregularities can be seen which make the film discontinuous and rougher, increasing the number of sites available for water vapors [10]. This discontinuity and roughness make the AAq thin film more favorable for humidity sensor. In our previous work on MO [27], the SEM micrographs were presented which revealed that roughness of the MO film leads to better humidity sensitivity of the sensor. Also, CoPc [29] and VOPc [10] morphological studies by using AFM analysis exhibited the same behavior, i.e., roughness was favorable for the humidity sensing of the sensor.

### 3.2. Device Characterization

The effect of relative humidity (%RH) on the capacitance and resistance (Figure 7) of the device is shown at frequencies 1 kHz and 120 Hz. To study the effect of relative humidity (%RH) on the capacitance or resistance of the sensor, the sensor was kept in a self-made glass chamber. The fabricated sensor was active between 40%RH–88%RH (the most active range in daily life) as capacitance increases and resistance decreases with increasing %RH. The capacitance and resistance of the sensor does not change dramatically from 0–40%RH because the number of vapors is inadequate to raise effective dielectric constant of the sensing material. Still, the capacitance increases significantly beyond 40%RH, which can be explained by the following equation:(1)$$C=\frac{\varepsilon_{o}\varepsilon_{r}A}{d}$$
where *C* is the capacitance of the sensor, *d* is separation/gap between capacitor plates, *A* is the active sensing area of the sensor, $\varepsilon_{o}$ is free space permittivity and $\varepsilon_{r}$ is relative permittivity of the material, which is 80 for water [36,37]. Although air is also present while measuring capacitance of the sensor; however, the relative permittivity of air is 1.00059 ~ 1 which is much smaller and negligible as compared to that of water, therefore, the major contribution in increasing the capacitance comes from water. At 120 Hz, net increase in capacitance is 19.2 pF, i.e., capacitance increase from 11.8 to 31 PF (which >2 fold), while at 1 kHz net change in capacitance is 2.8 pF (which is ~1.2 fold). The measurements are taken at two different frequencies (120 Hz and 1 kHz) to examine the performance of the sensor at low and high frequencies. Our device shows best result at lower frequency, i.e., 120 Hz according to the relation [38]
(2)$$f=\frac{1}{2\pi RC}$$

Surface irregularity and porosity of the organic semiconducting material layer of the device account for the rise in capacitance with rise in %RH. The more irregular and discontinuous the sensing layer, more water vapor will be absorbed which will cause to increase capacitance of the device. From SEM and AFM images, it is clear that surface of the AAq thin film is rough, irregular, and discontinuous so more water vapors are absorbed easily. The dielectric constant of the organic semiconducting materials lies in the range of 4–8 [39] whereas dielectric constant of water is 80.4 [36,37]. As a result, the variance between dielectric constants of organic semiconductor and water is very high leading to a large change in the capacitance of the sensor with the increase in dielectric constant of the surface. When the sensor is placed in a moist atmosphere, the active layer of the sensor absorbs water vapors in its pores, i.e., trapped in the pores which causes formation of two layers on the surface of the sensor, initially through chemisorption, and then a physisorption layer is formed. At lower humidity levels, chemisorption layer is formed while at advanced humidity levels, physisorption layer is also formed rapidly. When the %RH increases beyond 88RH%, the pores in the sensing material surface are completely filled and hence no increase in capacitance occurs as a result of saturation.

Sensitivity of the sensor can be calculated by using the following formula [40]:(3)$$S=\frac{C_{88}-C_{40}}{88-40}(pF/\%RH)$$

A linear approach was used in the %RH range from 40% to 88%RH, where *C*_88_ and *C*_40_ are the capacitance measured at 88%RH and 40%RH, respectively. The sensitivity of the sensor in the sensitive region was found to be 0.4 *p*F/%RH at 120 Hz and 0.06584 *p*F/%RH at 1 kHz.

It is also clear from the Figures that as %RH increases; the resistance decreases. At 120 Hz, resistance changes from 1.12 MΩ to 0.43 MΩ, while at 1 kHz resistance change from 0.13 MΩ to 0.10 MΩ. Decrease in resistance occurs as a result of the increase in conductance of the sensing material because when the %RH increases the sensing material absorb moisture which cause to increase conductance and as result decrease in resistance is observed. 

The resistance of a sensor depends on two factors: sensing material and it shape (geometry). This statement can be explained on the following equation: (4)$$R=\rho \frac{L}{A}$$
where *ρ* is the resistivity of the sensing material, *L* is length and *A* is area (geometry) of sensing layer. A small hysteresis is observed both in capacitance and resistance which is attributable the delayed desorption of water vapors comparatively to absorption. The small gap between hysteresis curves exhibits reproducibility as well as repeatability of the sensing measurements and stability of the device. 

Figure 8 shows the relation between temperature against capacitance at frequencies 120 Hz and 1 kHz respectively in temperature range from 295 K to 398 K at 30%RH. It is clear that capacitance change exponentially from 295 K to 393 K. At 120 Hz, the capacitance changes from 11.8 *p*F to 93 *p*F, which is 7.8 fold, increase while at 1 kHz, capacitance changes from 11.8 *p*F to 19.3 *p*F, or a 1.6-fold increase. The net change in capacitance at 120 Hz is 81.2 *p*F while at 1 kHz 7.5 pF. 

The temperature sensing mechanism for an AAq-based sensor can be explained from the effect of temperature on conductivity and mobility of AAq thin film. By increasing the temperature of the material, the resistivity decreases according to the relation
(5)$$\rho =\rho_{o}\exp\left(\frac{E}{kT}\right)$$
where *ρ* is the resistivity of sensing material at absolute temperature *T*, *ρ_o_* the pre-exponential factor and *E* is activation energy for conduction while *k* is Boltzmann constant. Also the conductivity and resistivity of a material are inversely related as;
(6)$$\rho =\frac{1}{\sigma }$$

This decrease in resistivity and increase in conductivity leads towards decrease in resistance of the device at a given frequency; as can be seen in Figure 8a,b. The capacitance of the sensor increases, as the resistance decreases with temperature according to Equation (2) that implies;
(7)$$C=\frac{1}{2\pi fR}$$

Therefore, the increase in capacitance of Au/AAq/Au sensor as a function of temperature is attributable to the decrease in resistance at a given frequency.

Sensitivity of temperature sensor can be calculated by using the following formula
(8)$$S=\frac{C_{388}-C_{295}}{388-295}(pF/T_{K})$$
where *S* is sensitivity, C*_295_* is capacitance at 295 K and C_388_ is capacitance at 388 K. The temperature sensitivity in this temperature region was found to be 0.87 pF/K at 120 Hz and 0.08 pF/K at 1 kHz respectively. As the temperature of organic semiconductor increases it give rise to conductivity and as a result, resistance decreases. The resistance decreases from 1.12 MΩ to 0.142 MΩ at 120 Hz and at 1 kHz resistance decreases from 0.134 MΩ to 0.082 MΩ. 

Several repeated measurements were taken both in forward and in reverse temperature and no appreciable hysteresis was observed which shows good stability and repeatability of device in response to temperature.

The sensor’s response and recovery times were measured by inserting the sensor from 30% to 90%RH and then then from 90% to 30%RH quickly. This process was performed at room temperature. The average value of the response time was measured to be 19 ± 2 s and the average value of recovery time was measured to be 25 ± 2 s. 

The overall sensing of our fabricated sensor is summarized in Table 1 in comparison to other reported sensors. 

## 4. Conclusions 

Herein, an organic semiconductor AAq is synthesized, characterized, and used in the fabrication of a surface type humidity and temperature sensor. Vacuum thermal evaporation is used to fabricate an Au/AAq/Au surface type sensor. Morphological studies indicate a rough surface favoring sensing applications. By investigating the sensing properties of a surface type Au/AAq/Au sensor, it was observed that capacitance and resistance varies with the change in either relative humidity or temperature. The capacitive and resistive response was studied as a function of relative humidity and temperature at two different frequencies, i.e., 120 Hz and 1 kHz. It was noted that at 120 Hz, the sensor was sensitive in comparison with 1 kHz. Therefore, at low frequencies the fabricated Au/AAq/Au sensor is favorable for its practical use either as humidity sensor or temperature sensor. 

## Figures and Tables

**Figure 1 materials-13-00960-f001:**
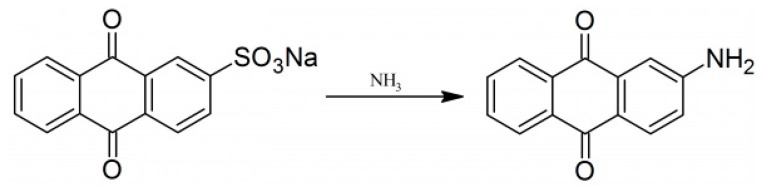
Cross synthesis of 2-amino anthraquinone (AAq).

**Figure 2 materials-13-00960-f002:**
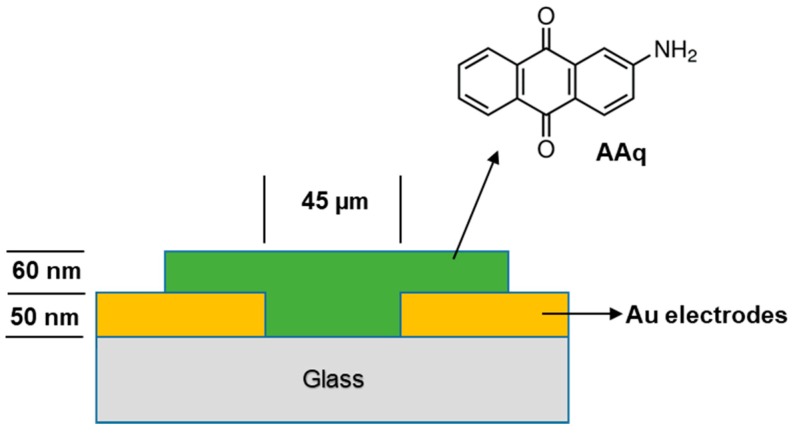
Cross sectional view of the Au/AAq/Au device showing the structure of AAq.

**Figure 3 materials-13-00960-f003:**
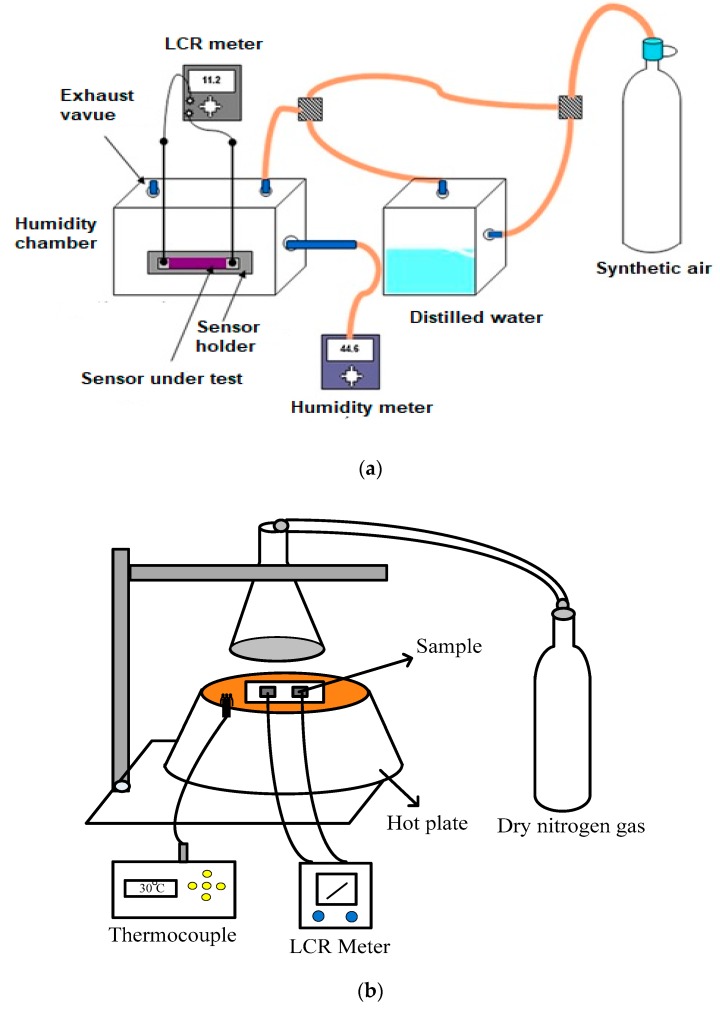
(**a**) Humidity measurement setup and (**b**) Temperature measurement setup used in this research.

**Figure 4 materials-13-00960-f004:**
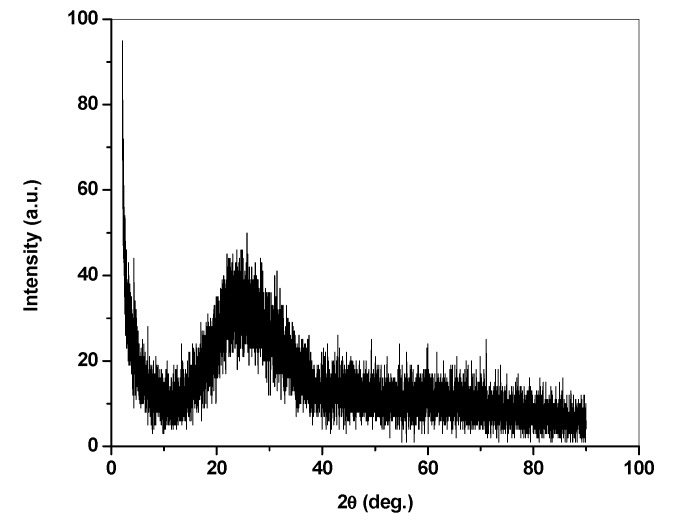
XRD pattern of AAq thin film thermally evaporated on glass substrate.

**Figure 5 materials-13-00960-f005:**
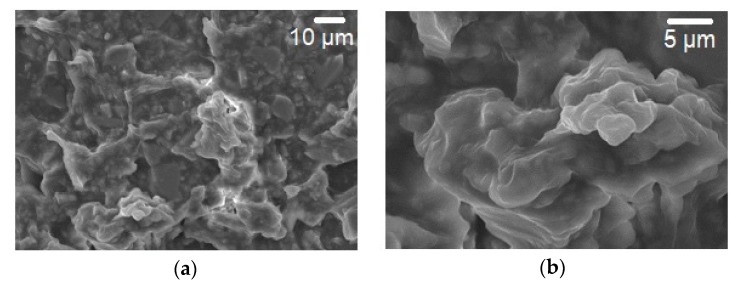
Scanning electron microscopy images (**a**) surface morphology of AAq thin film and (**b**) morphology of agglomerates.

**Figure 6 materials-13-00960-f006:**
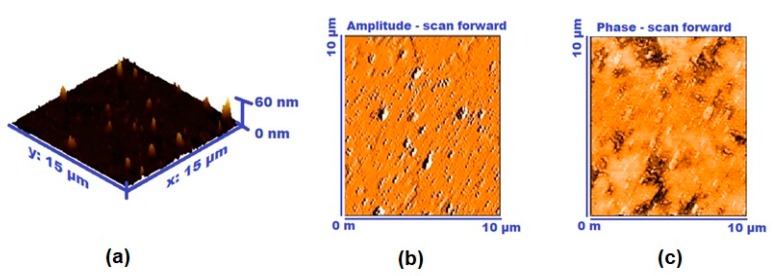
AFM images of (**a**) height analysis, (**b**) the coated glass substrate, and (**c**) phase analysis of AAq thin film.

**Figure 7 materials-13-00960-f007:**
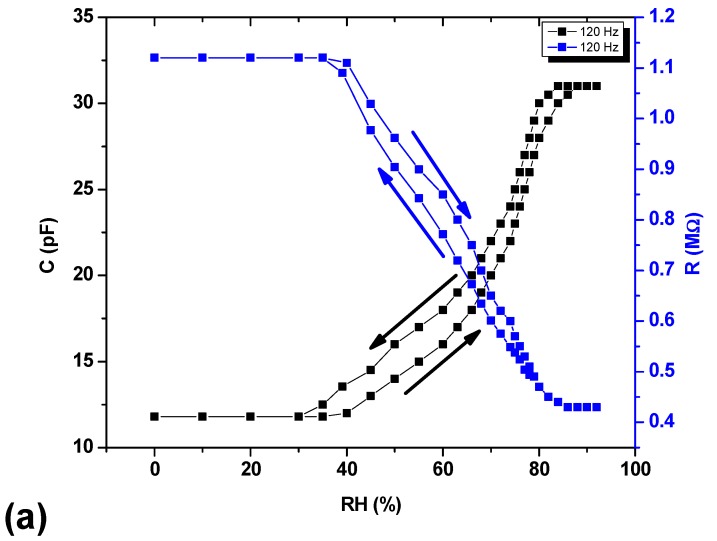
Effect of %RH on capacitance and resistance of the Au/AAq/Au sensor at (**a**) 120 Hz and (**b**) 1 kHz.

**Figure 8 materials-13-00960-f008:**
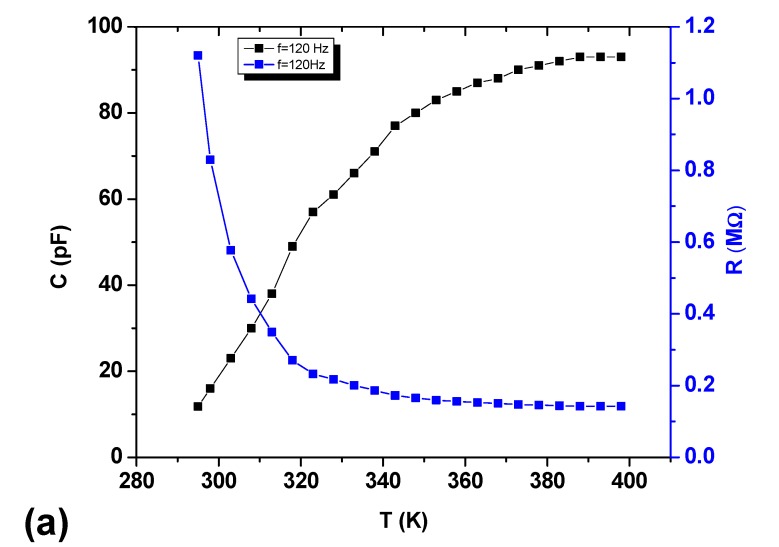
Effect of Temperature on Capacitance and Resistance at (**a**) 120 Hz and (**b**) 1 kHz.

**Table 1 materials-13-00960-t001:** Comparison of sensitivity and bandwidth of various devices with respect to relative humidity and temperature.

Devices	Relative Humidity		Temperature		Reference
	Sensitivity (pF/%RH)	Bandwidth (%RH)	Sensitivity (pF/K)	Bandwidth (K)	
Ag/MO/Ag	-	-	0.2	333−473	[32]
Au/CoPc/Au	8.2	65–93	0.4	330−462	[34]
Ag/CuTIPP/Ag	34	44–92	0.2	298−423	[33]
Au/AAq/Au	0.4	40–88	0.87	295−388	Present work

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
