# Peer review of "Amino Anthraquinone: Synthesis, Characterization, and Its Application as an Active Material in Environmental Sensors"

_materials, 2020, doi:10.3390/ma13040960_

Round 1

Reviewer 1 Report

In this manuscript, Ali and coworkers propose to investigate on using an anthraquinone derivative as new sensitive material for both temperature and humidity sensing via both resistive and capacitive transduction. The topic is interesting but the manuscripts lacks of information on important details that would increase the quality of the work before being published:

The introduction is focused on organic electronics, with humidity and temperature sensors (among other sensing platforms) being two applications for organic electronics.

I suggest the authors to tell about the advantages of organic electronic materials for humidity and temperature sensing, among the other technologies which already exist for these two different applications (with statements of these alternative technologies). The authors cite briefly 4 other organic molecules which have already been investigated in this field. Could the authors also state their advantages and drawbacks which motivated them to investigate on researching for anther molecular system?

Since no chemical analysis has been provided with the synthesis, can the author mention at what temperature AAq sublimes under reduced pressure (what was the based pressure for the evaporation/sublimation)? I assume that the sulfonate precursor does not sublime due to its ionic nature. Can the authors comment on that in the text in order to comfort that presumably only the product is evaporated on their substrate?

The material characterization which was provided is much appreciated. But, the authors commented only briefly (XRD+SEM+AFM on 12 lines of text only) on the observed morphology and on how the roughness will improve the humidity sensing. Can the authors comments a bit more what morphological properties are usually observed for the 4 molecules cited in the introduction?

Can the authors explain in the text on why they picked 1kHz and 120Hz?

In line 155, the authors use the permittivity of water and not the permittivity of air. This implies that the medium in which the capacitance is measured is water, which is not the case (since AAq is hydrophobic as mentioned line 31, liquid water shall not condense on AAq but the glass or the walls of the chamber). Surely the condensation of water with high permittivity may be a source of increase in the capacitance, but then the authors should have observed a increase with the humidity percent up to C(device, 100%) = 80*C(device, 0%). Here in the results, it is not the case, as the C increase by less than 3 at 120Hz, and less than 0.5 at 1 kHz. Such low effects can be due to surface changes. The authors are invited to discuss about it and revise their interpretation (lines 166-173) of the physical model in light of the amplitude of the measured effect and on possible alternate mechanisms.

In lines 187-189, the authors justify on the observed hysteresis as a delayed physical phenomenon. Can they mention about the speed for the RH variation? Could it be due to material degradation? Please comment in the text on this possible explanation.

On the temperature sensitivity, the authors have not displayed the back trace as they did for the humidity. Can the authors comment on the stability of the material’s performances with temperature? Has it been performed in air at atmospheric pressure (From Fig 3, it does not suggest that the temperature tests were necessarily performed in the controlled atmosphere chamber)?

On the possible mechanism of the temperature sensitivity (lines 202-206), the authors state that because the conductivity increase, the capacitance increase. These two properties are not correlated and the authors are invited to develop further.

In the conclusion, the authors summarize but do not explain on how to use their sensor: since it is sensitive to both RH and T, how to interpret if the measured output perturbations are due to either RH or T? This is not straightforward and the authors shall evoke perspectives. A solution might lie on multivariate analysis via dual frequency sensing as recently mentioned in the organic semiconductor sensing community for discrimination of both RH and T (by considering both 120Hz and 1kHz responses).

Reviewer 2 Report

This manuscript presents the synthesis of Amino-anthraquinone organic molecules, the characterization of thin films with X-ray diffraction and atomic force microscopy. Finally this organic semiconductor is embedded in a device and tested as a humidity sensor.

The literacy of the manuscript is generally very good, and the reading is pleasant. However, I have some serious concerns about the scientific interpretation of the experiment and the validity of some statements as discussed below.

1- The device fabricated is a planar metal/semiconductor/metal structure with 45 um separation between the metal contacts. The authors measure the capacitive and resistive response of this structure as a function of the humidity in the atmosphere. They provide an interpretation of the capacitive response with a parallel plate capacitor, however this cannot be applied to the geomery of Figure 2. The parallel plate capacitor is a layered structure with the two armour plates facing each other, the geometry in Figure 2 is totally different. The authors will need to calculate the capacitance for the specific scenario depicted in Figure 2 and draw a set of conclusions based on the correct expressions of the capacitance.

2- The authors state "The temperature sensing mechanism for an AAq based sensor can be explained from the effect of temperature on conductivity and mobility of AAq thin film". Whilst mobility is generally a function of mobility, this statement cannot simply be applied to the structures of Figure 2. The authors need to fabricate a field effect transistor and show that their interpretation is indeed correct. In lack of a direct study of the temperature dependence of the mobility, any statement on the mobility remains a pure speculation and conclusions based on a speculation are not scientifically sound.

In conclusion, I recommend the authors to calculate the expression for the capacitance corresponding to the device structure of Figure 2 and provide a sound interpretation of their data based on the correct model. I also recommend the authors to fabricate and measure a field effect transistor to measure the charge carrier mobility and base any interpretation on sound physical evidence. I recommend major revision.
